# Preparation of Al-Containing ZSM-58 Zeolite Membranes Using Rapid Thermal Processing for CO_2_/CH_4_ Mixture Separation

**DOI:** 10.3390/membranes11080623

**Published:** 2021-08-13

**Authors:** Eiji Hayakawa, Shuji Himeno

**Affiliations:** Department of Science of Technology Innovation, Nagaoka University of Technology, 1603-1, Kamitomioka-cho, Nagaoka, Niigata 940-2188, Japan; ehayakawa@chemeng.osakafu-u.ac.jp

**Keywords:** DDR, ZSM-58, rapid thermal processing, zeolite membrane, CO_2_ separation

## Abstract

The synthesis of DDR-type zeolite membranes faces the problem of cracks that occur on the zeolite membrane due to differences in the thermal expansion coefficient between zeolite and the porous substrate during the detemplating process. In this study, Al-containing ZSM-58 zeolite membranes with DDR topology were prepared by rapid thermal processing (RTP), with the aim of developing a reproducible method for preparing DDR zeolite membrane without cracks. Moreover, we verified the influence of RTP before performing conventional thermal calcination (CTC) on ZSM-58 membranes with various silica-to-aluminum (Si/Al) molar ratios. Using the developed method, an Al-containing ZSM-58 membrane without cracks was obtained, along with complete template removal by RTP, and it had higher CO_2_/CH_4_ selectivity. An all-silica ZSM-58 membrane without cracks was obtained by only using the ozone detemplating method. ZSM-58 crystals and membranes with various Si/Al molar ratios were analyzed by using Fourier-transform infrared (FTIR) spectroscopy to confirm the effects of RTP treatment. Al-containing ZSM-58 zeolites had higher silanol concentrations than all-silica zeolites, confirming many silanol condensations by RTP. The condensation of silanol forms results in the formation of siloxane bonds and stronger resistance to thermal stress; therefore, RTP caused crack suppression in Al-containing ZSM-58 membranes. The results demonstrate that Al-containing ZSM-58 zeolite membranes with high CO_2_ permeance and CO_2_/CH_4_ selectivity and minimal cracking can be produced by using RTP.

## 1. Introduction

Zeolites have been investigated due to their unique crystal structures. They can be used in gas separation membranes, in which a porous substrate is covered by a thin zeolite layer. For example, DDR-type zeolites have pore sizes of 0.36 nm × 0.44 nm; therefore, their application as a membrane material has been studied for the separation of CO_2_ (0.33 nm) from natural and digestive gases [1,2,3,4,5,6,7,8].

However, producing these membranes is challenging. Various phases might be generated from the target zeolite when the zeolite is synthesized on a porous substrate. Moreover, the production of many zeolites requires organic structure-directing agents (SDAs), which eventually need to be removed from the framework using high temperatures, which generates cracks due to differences in the thermal expansion coefficients between zeolite and the porous substrate [9,10]. The thermal expansion coefficients of DDR and the porous Al_2_O_3_ support are +35.1 × 10^−6^/K (298–492 K) to −8.7 × 10^−6^/K (492–1185 K) and ~7.5 × 10^−6^/K, respectively [1,11]. In fact, some researchers have reported that cracks form on DDR-type zeolite membranes during high-temperature calcination [1,2]. Therefore, inhibiting the occurrence of cracks to obtain membranes with a high gas separation performance is necessary.

Possible approaches in suppressing crack formation include the ozone detemplate method and rapid thermal processing (RTP). The ozone detemplate method can remove SDAs at lower temperatures than the conventional thermal detemplate method. The ozone detemplate method has been applied not only to DDR-type zeolite (DD3R, ZSM-58) membranes but also to other zeolite membranes [1,3,4,8,12,13].

RTP is a preprocessing method in which the zeolite membrane is rapidly heated and then cooled, and cracks can be suppressed during high-temperature calcination by increasing the bonding strength among zeolites during this process. RTP has been applied to some membranes, such as CHA and MFI [12,13,14,15,16,17,18]. However, Wang et al. reported that RTP did not affect DD3R zeolite membranes because of the low concentration of surface silanols in all-silica zeolite [1]. On the other hand, ZSM-58 has the DDR topology of Al-containing zeolites and is reported to have a Si/Al molar ratio of 20–∞ [19]. Therefore, we speculated that RTP treatment would result in crack suppression of Al-containing DDR-type zeolite membranes due to the concentration of silanol in the zeolite increasing with the Al concentration [20], and ZSM-58 zeolite has DDR topology. Recently, we reported that all-silica ZSM-58 zeolites can be used to afford high-purity DDR-type zeolite membranes with CO_2_/CH_4_ and CO_2_/N_2_ separation performances equivalent to those previously reported for DD3R membranes [4]. In this study, we examined the synthesis conditions of Al-containing ZSM-58 zeolite membranes as well as the effect of RTP treatment on ZSM-58 membranes with various Si/Al molar ratios. We compared the all-silica and Al-containing ZSM-58 membranes prepared by using different detemplate methods in terms of their performance in CO_2_/CH_4_ gas mixture separation. Moreover, the effect of RTP treatment was confirmed by observing changes in the silanol concentration of ZSM-58 zeolite crystals and membranes at various calcination conditions by using Fourier-transform infrared (FTIR) spectroscopy.

## 2. Experimental Procedure

### 2.1. Materials

The following chemicals and materials were used: Ludox HS-30 colloidal silica (30 wt.%, Sigma-Aldrich, St. Louis, MO, USA), sodium hydroxide (97.0%, Wako, Osaka, Japan), potassium hydroxide (85%, Wako), sodium aluminate (Al/NaOH molar ratio = 0.77, Wako, Osaka, Japan), tropine (98%, Combi-Blocks, San Diego, CA, USA), methyl iodide (99.5%, Wako), and ethanol (99.5%, Wako, Osaka, Japan). In addition, CO_2_ (99%) and CH_4_ (99%) were used for the gas permeation test. The alumina support used was the same as in our previous study [4]. The alumina support was composed of the surface, middle, and support layers; zeolite crystals were mainly synthesized on the surface layer. ZSM-58 zeolite membranes were prepared on the outside surface of porous α-alumina tubes (porosity, 30–50%; outer diameter, 9.6 mm; inner diameter, 7.2 mm; length, 3 cm; NGK Insulators, Ltd., Nagoya, Japan). The thicknesses of the surface, middle, and support layers in the alumina support were 50, 100, and 1050 μm, respectively. Moreover, the pore diameters of the surface, middle, and support layers in the alumina support were 0.2, 0.9, and 2–10 μm, respectively.

### 2.2. Preparation of Methyltropinium Iodide (MTI) as a Structure-Directing Agent

Methyltropinium iodide (MTI) was used as the structure-directing agent (SDA) in the synthesis of ZSM-58 zeolite. MTI was prepared following the protocol reported in our previous study [4]. For the synthesis of MTI, 25.0 g tropine was first dissolved in 100 g ethanol. Next, 25.1 g methyl iodide was added to the solution, and the suspension was maintained under reflux for 72 h. After cooling, the as-made MTI was washed with ethanol and dried at 343 K.

### 2.3. Synthesis of ZSM-58 Seed Crystals for Membrane Preparation

The ZSM-58 seed crystals were prepared following the protocol reported in our previous study [4]. For the synthesis of ZSM-58 seed crystals, a precursor solution with a composition molar ratio of MTI:NaOH:SiO_2_:H_2_O = 0.25:0.33:1:40 was used. The precursor solution was prepared as follows: 6.4 g MTI was added to 42.9 g H_2_O (solution A); 9.97 mL 3 M NaOH and 18.2 g Ludox HS-30 were then added to solution A and stirred for 24 h (solution B). Solution B was then transferred into a Teflon-lined autoclave and maintained at 433 K for 72 h under rotational stirring. The as-synthesized ZSM-58 zeolite seed crystals were washed with DI water. After drying at 343 K overnight, ZSM-58 seed crystals were calcined at 973 K for 4 h at a ramp rate of 0.5 K/min under atmosphere. Figure 1 shows the SEM image and XRD profile of ZSM-58 seed crystals after calcination.

The calcined ZSM-58 zeolite particles were then milled, and the ball-milled particles were dispersed in deionized (DI) water. Consequently, a ZSM-58 seed slurry with a particle size distribution of 0.2 μm was prepared by using a centrifugation method.

### 2.4. Preparation of Al-Containing ZSM-58 Crystals to Determine Optimal Conditions for Membrane Synthesis

The ZSM-58 zeolite crystal was prepared under various conditions in order to determine the optimal conditions for synthesis of Al-containing ZSM-58 zeolite membranes.

The precursor gel of ZSM-58 crystals (C1-8) was prepared as follows: MTI, an alkali source (NaOH or KOH), and sodium aluminate were added to DI water (solution A). Ludox HS-30 colloidal silica and the alkali source were also added to DI water (solution B). The precursor gel of ZSM-58 crystals was obtained by mixing and stirring solutions A and B for 24 h at room temperature. The gel composition of all samples had a Si/Al molar ratio of 70 and a H_2_O/SiO_2_ molar ratio of 52. Moreover, ZSM-58 crystals were prepared with molar ratios of MTI/SiO_2_ and MeOH/SiO_2_ from 0.05 to 0.3.

Next, 0.1 wt.% of ZSM-58 seed crystals was added to the precursor gel. Then, the precursor gel was transferred to a Teflon-lined autoclave, and hydrothermal treatment was conducted at a constant temperature (413 or 423 K) and time (24–192 h). After the hydrothermal treatment, the Teflon-lined autoclave was quickly cooled using water. The as-made crystals were washed with ethanol and DI water by filtration cleaning and dried overnight at 343 K. The as-made crystals were washed with DI water and dried at 343 K overnight. Finally, the ZSM-58 zeolite crystals were calcined at 823 K for 4 h at a ramp rate of 0.5 K/min under a standard atmosphere.

### 2.5. Preparation of ZSM-58 Zeolite Membranes

ZSM-58 zeolite membranes were prepared on α-alumina tubular with the secondary growth method. Ball-milled ZSM-58 seed particulates were coated on the outermost surface layer of the alumina tubular support using a dip-coating method with ZSM-58 seed slurry.

The precursor gel for the Al-containing ZSM-58 membrane (M1-4) was prepared by using the method described in Section 2.4. The Al-containing ZSM-58 membranes were prepared from a gel composition of MTI:KOH:SiO_2_:H_2_O = 0.05:0.05:1:52 at 413 K for the requisite synthesis time. All-silica ZSM-58 membranes (M5 and M6) were prepared from a gel composition of MTI:KOH:SiO_2_:H_2_O = 0.1:0.1:1:52 at 413 K for 48 h, based on our previous report which investigated the influence of the membrane Si/Al molar ratio on RTP [4]. Please note that KOH was used for the synthesis of all-silica ZSM-58 membranes to mitigate the influence of the alkali source, unlike in our previous procedure. α-Alumina tubular support-coated ZSM-58 seed crystals were placed vertically in a Teflon-lined autoclave. A precursor gel was then transferred into the Teflon-lined autoclave, and hydrothermal treatment was conducted. Subsequently, the as-made membrane was washed with boiling water and dried overnight at 343 K.

### 2.6. Calcination of ZSM-58 Zeolite Membranes

The as-made ZSM-58 membranes were calcined using the conventional thermal calcination (CTC) method or ozone detemplate method. For some membranes, RTP was applied prior to the CTC method.

Figure 2 shows the temperature program of the RTP and CTC process. RTP has been reported in other zeolite membranes, such as MFI and CHA-type zeolites [12,13,14,15,16,17,18], and RTP treatment conditions were selected based on those reports. The membrane was directly loaded into a preheated muffle furnace at 973 K for 1 min and then quickly removed and cooled to room temperature under flowing air. Then, the CTC method was performed in which the membrane was heated at 823 K for 4 h at a heating and cooling rate of 0.5 K/min in a muffle furnace atmosphere.

The calcination conditions of the ozone detemplate method (ozone) were obtained from the literature [3,4,8,12,13]. Figure 3 shows the ozone detemplate apparatus. The ozone mixture was generated from oxygen gas. The zeolite layer was synthesized on the outer surface of the alumina tubular support, this zeolite surface layer was exposed to the oxygen–ozone gas mixture (O_2_ + O_3_) the. The membrane was placed on a membrane module in contact with ozone at a concentration of 120 g/Nm^3^ and a flow rate of 1 L/min (about 110 mL/min/cm^2^) at 473 K for 48 h. 

### 2.7. Gas Permeation and Separation Measurements

The pure CH_4_ gas permeance was measured for the noncalcined ZSM-58 zeolite membrane in order to evaluate its denseness immediately after hydrothermal synthesis; the feed pressure was 1.1 MPa. 

Moreover, an equimolar CO_2_/CH_4_ mixture, prepared by blending pure CO_2_ and CH_4_ gas, was used to evaluate the permeance characteristics of ZSM-58 zeolite membranes. The membrane was dried at 423 K prior to the permeation test. The total feed pressure of the CO_2_/CH_4_ mixture gas was 1.1 MPa, and the permeate pressure was atmospheric pressure (about 0.1013 MPa). The gas permeation temperature was maintained at 298 K, and the gas flux and concentration were determined with a soap film flowmeter and a gas chromatograph (490 Micro GC, GL Science, Tokyo, Japan), respectively. The permeation test was conducted by using a membrane that had dried under dry gas flow at 383 K.

The permeance, *P_i_* (mol/s/m^2^/Pa), was calculated using the following formula:(1)$$P_{i}=\frac{Q_{i}}{A\cdot t\cdot \Delta p_{i}}\;\;$$
where *Q_i_* (mol) is the mole of component *i* permeated through the membrane for *t* (s), *A* (m^2^) is the membrane area, and Δ*pi* (Pa) is the differential partial pressure of component *i* between the feed and permeate side. The CO_2_/CH_4_ separation selectivity (PCO_2_/PCH_4_) was the permeance ratio of the mixture and gas. The feed rate of the CO_2_/CH_4_ mixture was 1 L min^−1^, and the differences in gas concentrations between the feed and retentate sides were less than 1.0%. Therefore, we discounted the effect of concentration polarization in the gas mixture on CO_2_/CH_4_ selectivity.

### 2.8. Characterization

Nitrogen adsorption isotherms were measured at 77 K and with BELSORP-max apparatus (MicrotracBEL Corp, Osaka, Japan). The XRD (XRD-6100, Shimadzu, Kyoto, Japan) profiles were recorded by using Cu–Kα radiation. The SEM images were obtained on a field emission SEM (JSM-6701, JEOL Ltd., Tokyo, Japan) operated at an accelerating voltage of 1 kV and a work distance of 8–11 mm. Element mapping and line analysis were performed using energy-dispersive X-ray spectrometry (EDS). Elemental analysis of the ZSM-58 zeolite crystals was conducted by using X-ray fluorescence spectroscopy (XRF; Rigaku ZSX Primus II, Rigaku Co., Tokyo, Japan). FTIR spectroscopy was measured by JASCO FT/IR-4100(JASCO Corporation, Tokyo, Japan), and the FTIR spectra of ZSM-58 crystals and membranes were obtained by attenuated total reflectance (ATR) by using ATR PRO ONE attached to a ZnSe prism (JASCO Corporation, Tokyo, Japan).

## 3. Results and Discussion

### 3.1. Si/Al Molar Ratios and Crystallization Rates of ZSM-58 Crystals at Various Gel Compositions

Initially, the effects of the alkali source and gel composition on ZSM-58 crystals were investigated in order to determine the synthesis conditions for Al-containing ZSM-58 zeolite membranes. Table 1 shows the Si/Al molar ratios of ZSM-58 crystals by XRF with different gel compositions. All samples were synthesized for 2 d at 423 K. Compared with NaOH, KOH produced ZSM-58 crystals with a lower Si/Al molar ratio due to the high dissolution ability of the aluminosilicate [21]. In addition, the high MeOH/SiO_2_ molar ratio induced a low Si/Al molar ratio. Thus, KOH is the most suitable alkali to efficiently load Al into zeolites.

Subsequently, the crystallization rate of different gel compositions was compared with the rates observed when KOH was used as the Me source. Figure 4 shows the weight ratios of ZSM-58 to raw silica with different synthesis times. Figure 5 shows the XRD peaks of the as-fabricated C5–C8 crystals with various synthesis periods. The crystallization rate was high at a temperature of 423 K; therefore, this experiment was conducted at 413 K. After 2 d, C5 did not form a ZSM-58 zeolite and neither did C6–C8 after 1 d, as evidenced by XRD. Thus, the amount of synthesized ZSM-58 was taken as 0 g. The prepared powders obtained after longer synthesis times had high purity and crystallinity, confirmed from the XRD profiles (Figure 5); therefore, we assumed that the amount of synthesized ZSM-58 was the same as the weight of the as-prepared powder. For compositions C6–C8, the ZSM-58 zeolite rapidly crystallized within 24–48 h. By contrast, the ratio of the zeolite to raw silica increased with the synthesis time under C5 conditions for 48 h, suggesting that the rate of crystal nucleation under C5 conditions was slower. Slow crystal nucleation was observed in low MTI and SiO_2_ concentration; thus, these were assumed to be good crystal growth conditions. When synthesizing a dense zeolite membrane, the grain boundaries must be filled by growing seed crystals in a porous α-alumina support; therefore, it was concluded that the conditions for C5 are the most suitable of those tested for synthesizing Al-containing ZSM-58 zeolite membranes.

### 3.2. Influence of Synthesis Time on Al-Containing ZSM-58 Membranes

As shown in Table 2, the influence of the synthesis time on the separation performance of Al-containing ZSM-58 zeolite membranes was monitored by using composition C5 at 413 K. M1–M4 membranes were calcined via the CTC method, and RTP was applied prior to CTC. Figure 6 and Figure 7 show the XRD profiles and SEM images of the surfaces and cross-sections of M1–M4, respectively. Figure 8 shows the CO_2_ permeance and CO_2_/CH_4_ selectivity of Al-containing ZSM-58 membranes as functions of synthesis time; each value are the averages of two samples prepared by same synthesis condition. The maximum error of CO_2_ permeance between the two samples is 1.0 × 10^−8^ mol/m^2^/s/Pa.

According to the XRD profile, the Al-containing membrane included a high-purity ZSM-58 phase (Figure 6). Moreover, the membrane thickness tended to increase as the synthesis time increased (Figure 7). Therefore, the CO_2_ permeance of the Al-containing ZSM-58 membrane decreased as a function of synthesis time because (Figure 8). We measured the CH_4_ permeance of non-calcined ZSM-58 membranes in order to evaluate its denseness immediately after hydrothermal synthesis. The growth of zeolite crystals and the decrease in CH_4_ permeance of non-calcined ZSM-58 membrane were promoted with the increase in synthesis time. These results suggest that for the low-MTI and -KOH compositions, such as C5, the zeolite layer densification was promoted by the growth of zeolite crystals on the alumina substrate. This is demonstrated by the tendency of the CO_2/_CH_4_ selectivity of Al-containing ZSM-58 membranes to increase as a function of synthesis time. The CO_2_ permeance of M4 was slightly lower than those of M1 and M2 due to dense zeolite layer. However, the dense zeolite layer of M4 caused very low CH_4_ permeance; therefore, M4 had the highest CO_2_/CH_4_ selectivity among the results, as presented in Table 2. Therefore, a dense Al-containing ZSM-58 membrane with few defects was prepared after a long growing time in the calcined RTP and CTC process, which showed high CO_2_ permeance and CO_2_/CH_4_ selectivity.

### 3.3. Membrane Structure of Al-Containing ZSM-58 Membranes

Figure 9 shows the SEM and EDS results of the Al-containing ZSM-58 membrane (M4); the green and red mapping in the EDS mapping indicate Si and Al, respectively. The EDS analysis was performed on the surface layer of the alumina support; this confirmed that almost all synthesized zeolite crystals were on the surface layer of the alumina support. The Si/Al molar ratio of gel composition of ZSM-58 zeolite was 70, and ZSM-58 zeolite on the support mainly comprised Si. Thus, the Si and Al in EDS mapping correspond to the ZSM-58 zeolite and alumina support, respectively.

The structure of the M4 membrane was classified into layers A, B, and C using line analysis. Layer A comprised only the ZSM-58 zeolite layer due to a few Al intensities. The thickness of layer A was approximately 2 μm, and the SEM image of layer A confirmed that ZSM-58 crystals were synthesized outside the alumina support. Moreover, because Si mapping was present in the alumina support, the ZSM-58 zeolite was widely synthesized in the alumina surface layer. Layer B had a dense zeolite/alumina composite layer with a thickness of approximately 6 μm, because the Si intensity was constant and Al intensity gently increased. It was confirmed by SEM that the ZSM-58 zeolite in layer B was densely synthesized in porous alumina. By contrast, layer C had a low Si/Al intensity ratio, and vacancies in the alumina support were observed via SEM. Therefore, the gas permeation resistance was larger in layer B than in layer C, and layer C did not contribute to gas separation.

The DD3R membrane calcined by the ozone detemplate had approximately 1340 barrer of CO_2_ permeability (CO_2_ permeance, 3 × 10^−7^ mol/m^2^/s/Pa; thickness, 1.5 μm) [1]; it was considered that the CO_2_ permeability of the ZSM-58 membrane was similar to that of the DD3R membrane because ZSM-58 had the same crystal structure as DD3R. M4 was approximately 1900 barrer of the CO_2_ permeability calculated by using the thicknesses of layer A and B (8 μm), and it had higher CO_2_ permeability than the DD3R membrane calcined by the ozone detemplate because molecular MTI was almost removed from the zeolite framework in M4 due to the RTP and CTC processes. Therefore, M4 was configured from the thin outer surface zeolite layer (layer A) and dense zeolite/alumina composite layer (layer B), which showed high CO_2_ permeability after applying RTP and CTC processes.

### 3.4. Comparison with All-Silica ZSM-58 Zeolite Membranes

We compared the CO_2_ separation performance (CO_2_ permeance; CO_2_/CH_4_ selectivity) of all-silica ZSM-58 membranes by employing various calcination methods in order to verify the effect of RTP treatments on Al-containing ZSM-58 membrane. Table 3 shows the CO_2_/CH_4_ separation performance of all-silica ZSM-58 membranes (M5 and M6) with different calcination methods. M5 and M6 were prepared from a gel composition of MTI/KOH/SiO_2_/H_2_O = 0.1:0.1:1:52 at 413 K for 48 h, based on our previous study [4]. Notably, KOH was used to synthesize all-silica ZSM-58 membranes in order to eliminate the effect of an alkali source, unlike in our previous procedure. Figure 10 shows the SEM images of M1 and M5. The weight loss after calcination indicates the removal rate of the template in the zeolite framework and was calculated by using the following equation:(2)$$Weight\;loss\;\left(\%\right)=\left(1-\frac{W_{mem,cal}-W_{support}}{W_{mem}-W_{support}}\;\right)\times 100\;\;$$
where *W_mem_*, *W_mem,cal_*, and *W_support_* (g) are the weights of the as-fabricated membrane, membrane after calcination, and α-alumina support, respectively. The ZSM-58 zeolite contained six cages with eight-membered windows per unit cell, and this unit cell comprised Si_120−X_Al_X_O_240_. The theoretical weight loss is approximately 11.5% if the MTI molecule were to be completely removed from the zeolite pore, assuming that the cage with an eight-membered window contains one MTI molecule.

M5 and M6 had a dense ZSM-58 membrane with low CH_4_ permeance before calcination. However, the CH_4_ permeance of M5 was increased following the RTP and CTC processes. As shown Table 2, M5 had higher CH_4_ permeance and lower CO_2_/CH_4_ selectivity than the Al-containing ZSM-58 membrane with a similar membrane thickness (M1). This indicates that, for the all-silica ZSM-58 membrane, the RTP treatment did not exhibit crack suppression effects. RTP strengthens grain boundaries because of the condensation of surface silanol groups (Si–OH) [15]. However, it was reported by Wang et al. that RTP had no crack suppression effects on all-silica DDR membranes because the all-silica zeolite membrane has few surface silanols [1]. The Al-containing ZSM-58 membrane (M1) had many surface silanol groups compared to the all-silica ZSM-58 membrane (M5); therefore, the formation of cracks appeared to be suppressed by applying the RTP treatment. 

Moreover, an all-silica ZSM-58 membrane with improved CO_2_/CH_4_ separation performance could be prepared using the ozone detemplate method [4]. Crack generation was suppressed because the ozone detemplate method could remove the template molecule from zeolite at a lower temperature than the CTC method. M6 was calcined without cracks by the ozone detemplate method, and its CO_2_/CH_4_ separation selectivity showed a similar value to that of M4. Although the all-silica membrane had a thinner zeolite layer than the Al-containing ZSM-58 membrane (Figure 10), M6 had a CO_2_ permeance similar to that of M4. Therefore, the CO_2_ permeability of M6 was approximately 790 barrer, which was lower than that of M4 (1900 barrer). This was due to insufficient detemplating from the zeolite framework. The M6 membrane could only decrease in weight by about half of the theoretical values, and the ozone detemplate method could not completely remove the template, unlike thermal calcination [12]. The molecular template in the zeolite framework inhibited the diffusion of molecular CO_2_ into zeolite pores. On the other hand, all samples (M1–4) in Table 2 had a weight loss of approximately 11%; RTP and CTC processes could eliminate molecular MTI from the zeolite framework. 

In summary, RTP treatment could be utilized to prepare Al-containing zeolite membranes without cracks. In addition, almost all molecular templates could be eliminated from the zeolite framework during the CTC process. Therefore, RTP treatment is effective for preparing Al-containing ZSM-58 membranes with high CO_2_ permeability and CO_2_/CH_4_ selectivity.

### 3.5. Characteristics of ZSM-58 Crystals and Membranes with Different Si/Al Molar Ratios and Calcination Conditions by FTIR

It is considered that the RTP treatment suppressed cracks in the Al-containing zeolite membrane with a high silanol concentration due to surface silanol condensation (Figure 11a) [1,15]. Therefore, we investigated the effect of RTP treatment by monitoring the silanol concentration of ZSM-58 zeolite crystals and membranes with various Si/Al molar ratios as the calcination process progressed. The surface silanol concentration was measured by FTIR, as per a report on silica membranes [22,23,24], and the silanol concentration was evaluated from the peak area ratio of surface silanol groups (Si–OH) to siloxane (Si–O–Si) (Figure 11b). The surface silanol group (Si–OH) was attributed to peaks at 960 cm^−1^, and the siloxane (Si–O–Si) groups were attributed to peaks at 1060, 1100, 1170, and 1210 cm^−1^ [23]. The FTIR spectra were deconvoluted by using the Fityk program based on Gaussian peaks [25], and the area of each peak was measured. 

Figure 12 shows the FTIR spectra of the ZSM-58 zeolite crystals with various Si/Al molar ratios between 500 and 1500 cm^−1^. Table 4 shows the synthesis conditions and peak area ratios of Si–O–H/Si–O–Si of the ZSM-58 crystals with different Si/Al molar ratios. The silanol concentration of the ZSM-58 crystal before RTP treatment increased proportionally to the Al concentration (Table 4). However, the silanol concentrations of ZSM-58 crystals were decreased by RTP treatment, and the decrease in silanol concentration indicated the formation of siloxane bridges due the condensation of surface silanol groups. Figure 13 shows the degree of diminution of the peak area ratio of Si–OH/Si–O–Si on the ZSM-58 crystal before and after RTP treatment. The ZSM-58 zeolite crystals with lower Si/Al molar ratios (Si/Al = 40 and 70) had greater decreases in the peak area ratio of Si–OH/Si–O–Si than that of the high-silica ZSM-58 zeolite crystals (Si/Al = 300 and ∞). This result suggests that more siloxane bridges were formed by silanol condensation on Al-rich ZSM-58 zeolite during RTP treatment.

Next, we directly confirmed surface silanol group condensation from the FTIR spectrum of the ZSM-58 membrane. The FTIR spectrum of the ZSM-58 zeolite membrane was obtained by ATR [26]. Table 5 shows the peak area ratios of Si–OH/Si–O–Si for all-silica and Al-containing ZSM-58 zeolite membranes. These membranes were prepared by using the same conditions as in M4 and M5. Figure 14 shows the FTIR spectra of the Al-containing and all-silica ZSM-58 zeolite membranes before and after RTP treatment. Figure 15 shows the decrease in peak area ratio of Si–OH/Si–O–Si on the ZSM-58 membrane by RTP treatment. As with the ZSM-58 crystal, the Al-containing ZSM-58 zeolite membrane before RTP treatment had a higher silanol concentration and a greater decline in the peak area ratio of Si–OH/Si–O–Si after RTP treatment than the all-silica ZSM-58 zeolite membrane. These results suggest that the Al-containing membrane improves thermal stress resistance during the CTC process because siloxane bridges were formed from surface silanol groups on the zeolite membrane with high Al concentrations due to RTP treatment. In summary, we confirmed that a ZSM-58 membrane with high thermal resistance was formed because bonding between zeolite crystals was enhanced by forming siloxane bridges due to silanol condensation.

## 4. Conclusions

We studied the synthesis conditions for an Al-containing ZSM-58 zeolite membrane using RTP treatment. For the preparation of Al-containing and all-silica ZSM-58 zeolite membranes, RTP treatment was applied before the CTC process; as a result, the suppression effect of formed cracks was confirmed only in the Al-containing ZSM-58 zeolite membrane. The CTC method with RTP treatment completely removed the template and enhanced the CO_2_ permeance and CO_2_/CH_4_ selectivity of the Al-containing ZSM-58 membrane. The all-silica ZSM-58 membrane without cracks was prepared by only using the ozone detemplate method. In particular, the Al-containing ZSM-58 zeolite membrane with high CO_2_/CH_4_ separation selectivity was prepared by slow crystallization in solution, with a gel composition of MTI:KOH:SiO_2_:H_2_O = 0.05:0.05:1:52 at 413 K for 120 h.

The effect of RTP was confirmed by the FTIR analysis of ZSM-58 zeolite crystals and membranes prepared using different calcination methods. The surface silanol concentration increased with Al concentration in the zeolite, and siloxane bond formation on the ZSM-58 zeolite with a low Si/Al molar ratio occurred due to more surface silanol condensation following RTP. These trends were observed on both ZSM-58 zeolite crystals and membranes. Therefore, we demonstrated the effectiveness of RTP treatments in the preparation of Al-containing ZSM-58 membranes by monitoring the silanol concentrations of ZSM-58 zeolite membranes with different Si/Al molar ratios as the calcination process progressed. 

In summary, RTP treatment was the most effective approach to achieve efficient template removal and crack-free Al-containing ZSM-58 membranes, and we have presented a new method in this paper for the preparation of DDR-type zeolite membranes without cracks by applying RTP treatments to Al-containing ZSM-58 zeolite.

## Figures and Tables

**Figure 1 membranes-11-00623-f001:**
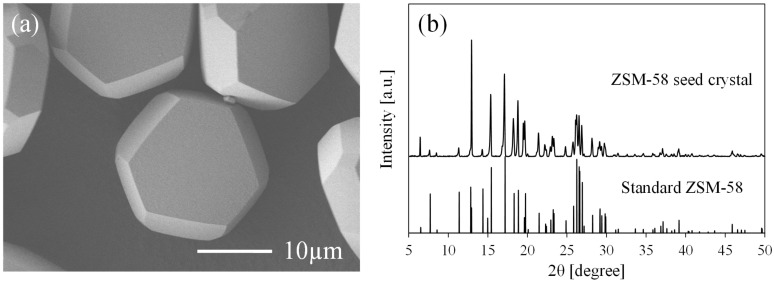
(**a**) SEM image and (**b**) XRD profile of ZSM-58 seed crystals used in this study.

**Figure 2 membranes-11-00623-f002:**
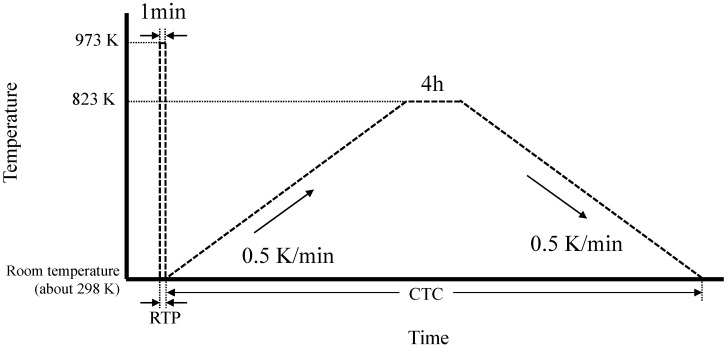
Temperature program of RTP and CTC.

**Figure 3 membranes-11-00623-f003:**
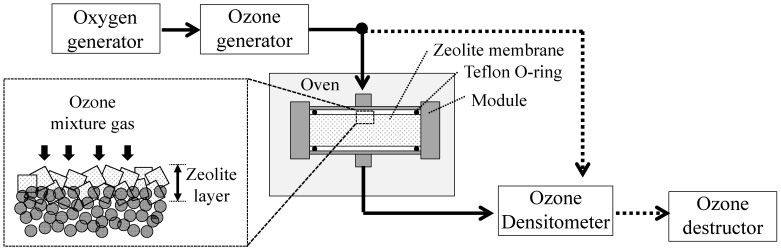
Schematic of the ozone detemplate apparatus.

**Figure 4 membranes-11-00623-f004:**
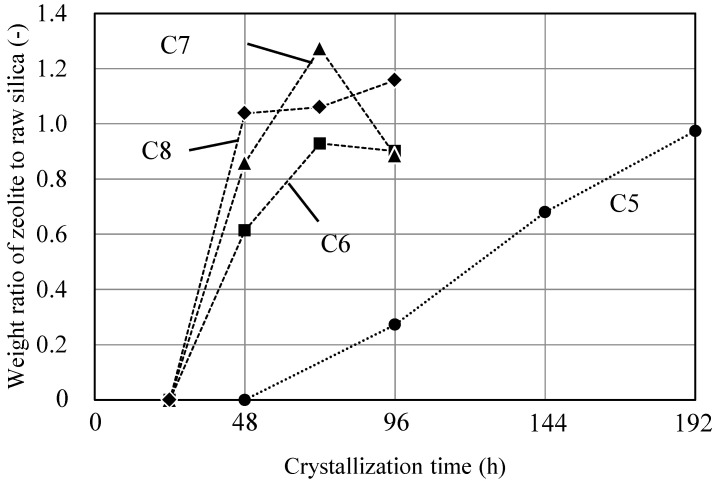
Weight ratio of zeolite to raw silica for gel compositions C5–C8 at 413 K.

**Figure 5 membranes-11-00623-f005:**
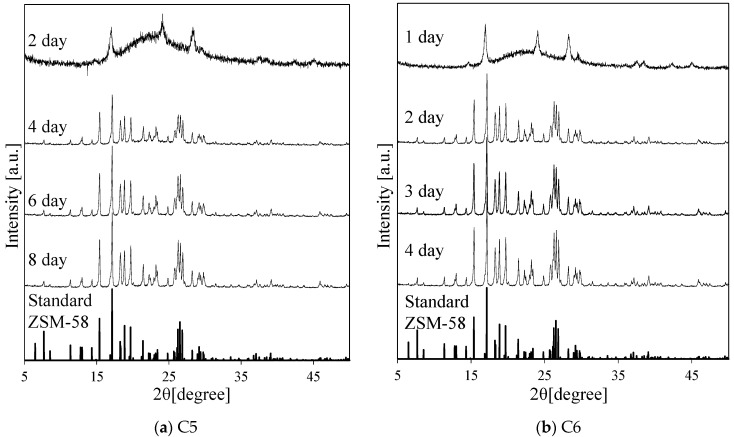
XRD profiles of the as-made crystals with various synthesis periods and gel compositions of (**a**) C5, (**b**) C6, (**c**) C7 and (**d**) C8 at 413 K.

**Figure 6 membranes-11-00623-f006:**
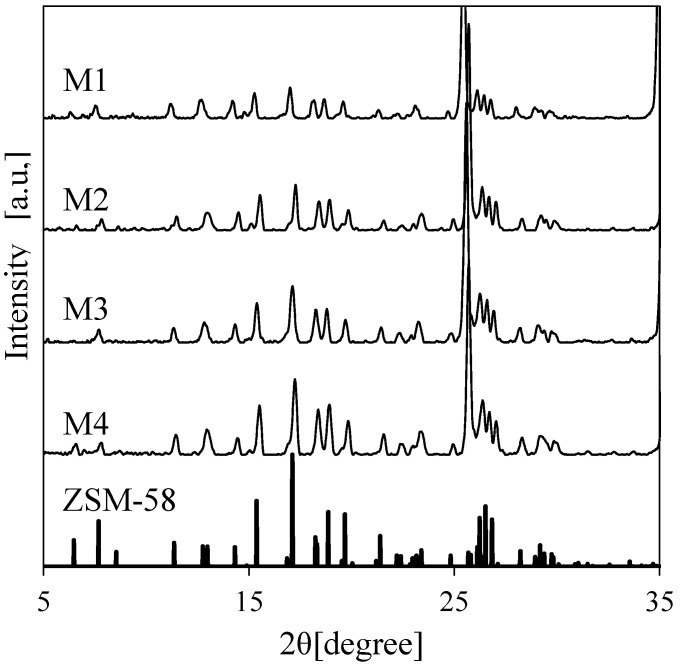
XRD profile of ZSM-58 zeolite membranes using various synthesis periods.

**Figure 7 membranes-11-00623-f007:**
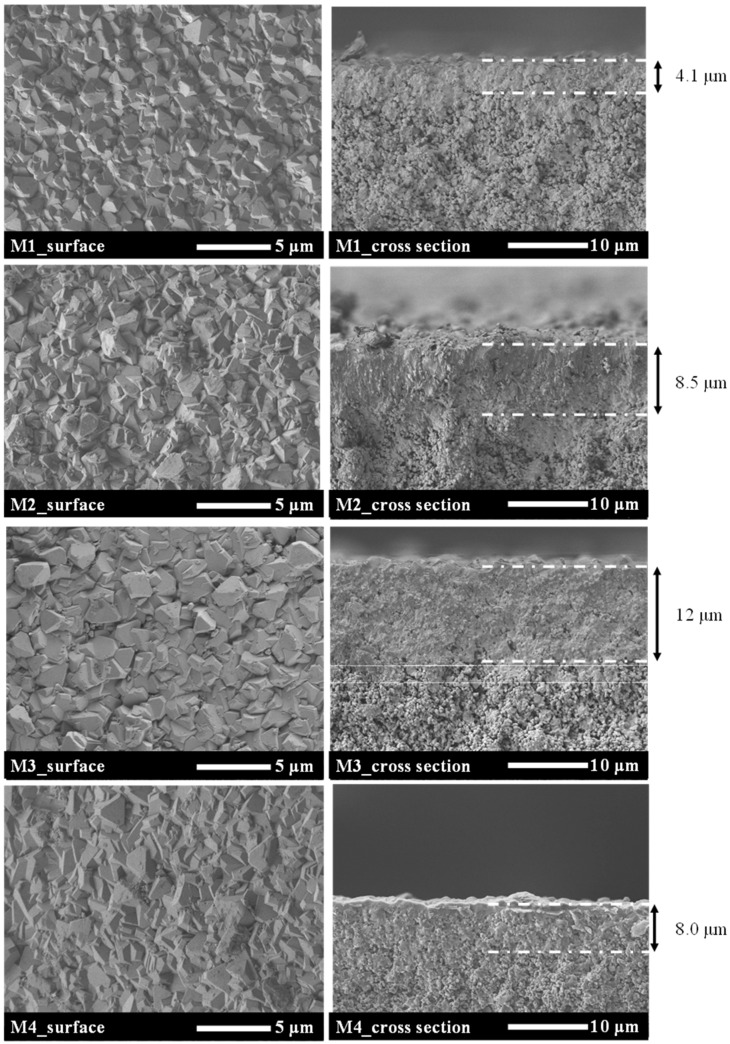
SEM images of the surfaces and cross-sections of Al-containing ZSM-58 zeolite membranes synthesized for various periods. The arrows denote the membrane thickness.

**Figure 8 membranes-11-00623-f008:**
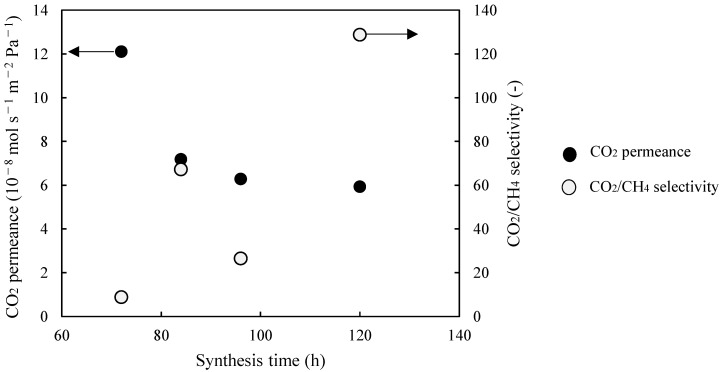
CO_2_ permeance and CO_2_/CH_4_ selectivity of Al-containing ZSM-58 membranes as functions of synthesis time. CO_2_ permeance and CO_2_/CH_4_ selectivity values are averaged from two samples.

**Figure 9 membranes-11-00623-f009:**
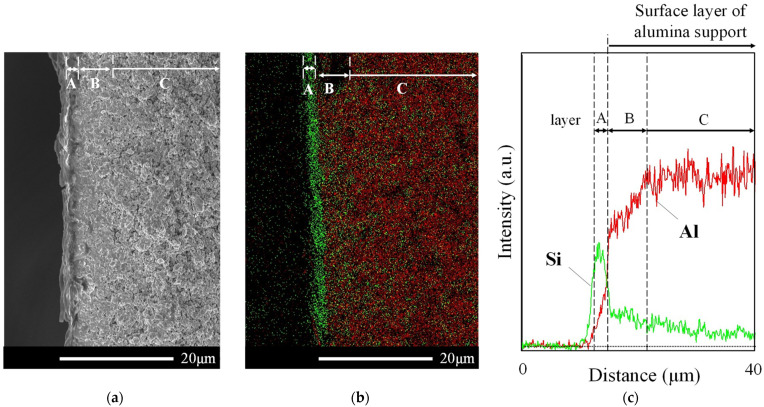
SEM image (**a**), element mapping (**b**), and line analysis (**c**) of the cross-section of the Al-containing ZSM-58 membrane (M4). Si and Al are indicated by green and red, respectively.

**Figure 10 membranes-11-00623-f010:**
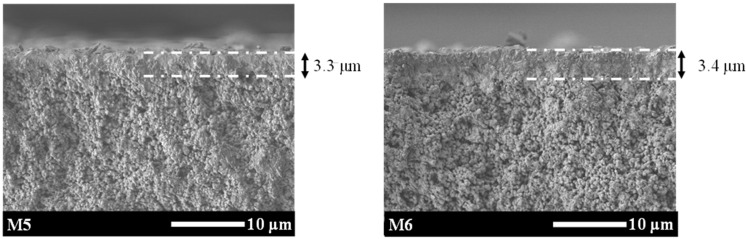
SEM of the cross-sections of all-silica ZSM-58 zeolite membranes M5 and M6. The arrows indicate the thickness of the zeolite membrane.

**Figure 11 membranes-11-00623-f011:**
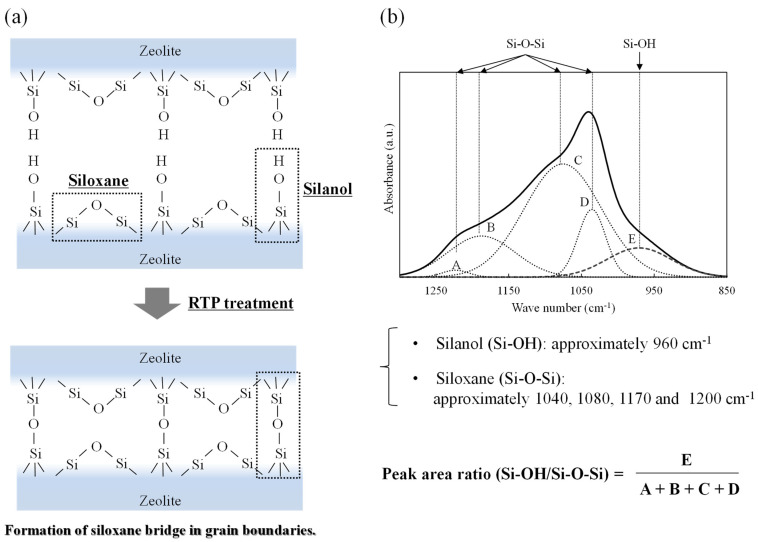
(**a**) Schematic formation of silanol bridges by silanol condensation, and (**b**) FTIR spectrum with deconvoluted peak components using the Gaussian line shape [1,15,23].

**Figure 12 membranes-11-00623-f012:**
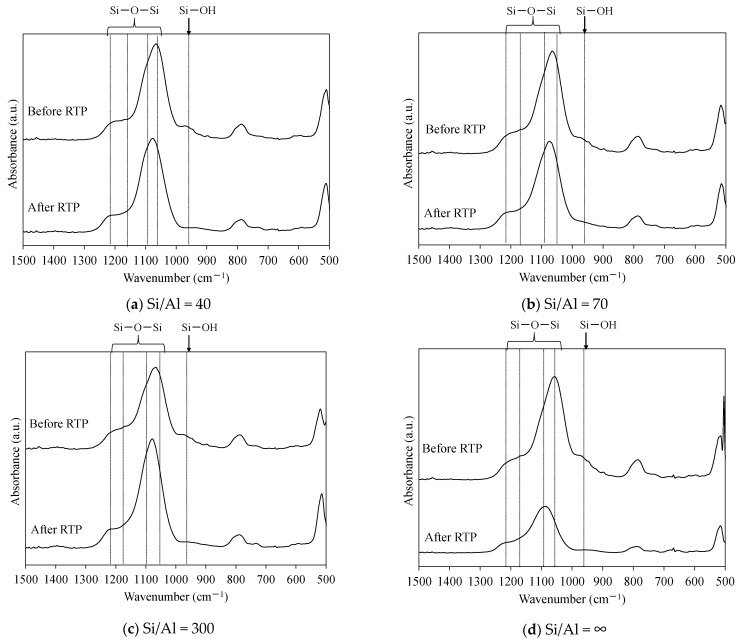
FTIR spectra (500 to 1500 cm^−1^) of the as-made ZSM-58 zeolite crystals with different Si/Al molar ratios (**a**–**d**) before and after RTP treatment.

**Figure 13 membranes-11-00623-f013:**
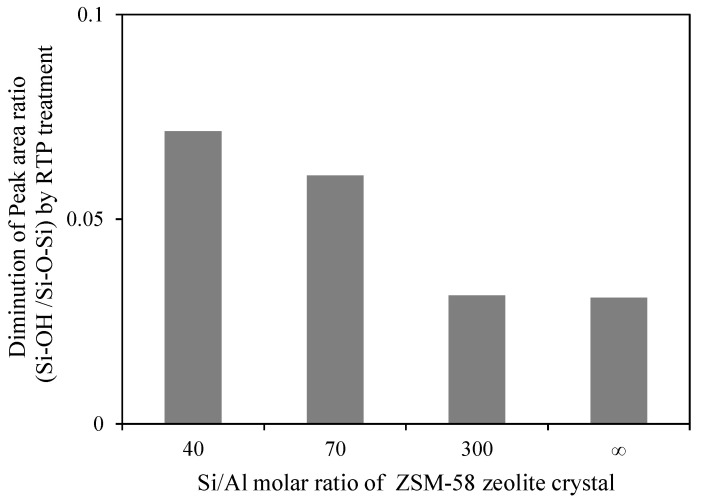
Diminution of peak area ratio (Si–O–H/Si–O–Si) on ZSM-58 crystals with different Si/Al molar ratios after RTP treatment.

**Figure 14 membranes-11-00623-f014:**
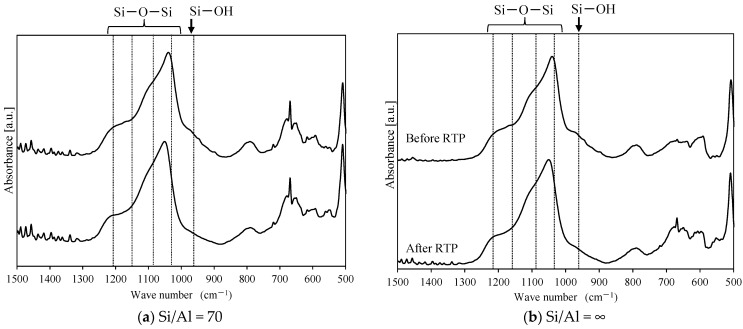
FTIR spectra (500 to 1500 cm^−1^) of ZSM-58 zeolite membranes with Si/Al ratios of 70 (**a**) and ∞ (**b**) prepared using the ATR method before and after RTP treatment.

**Figure 15 membranes-11-00623-f015:**
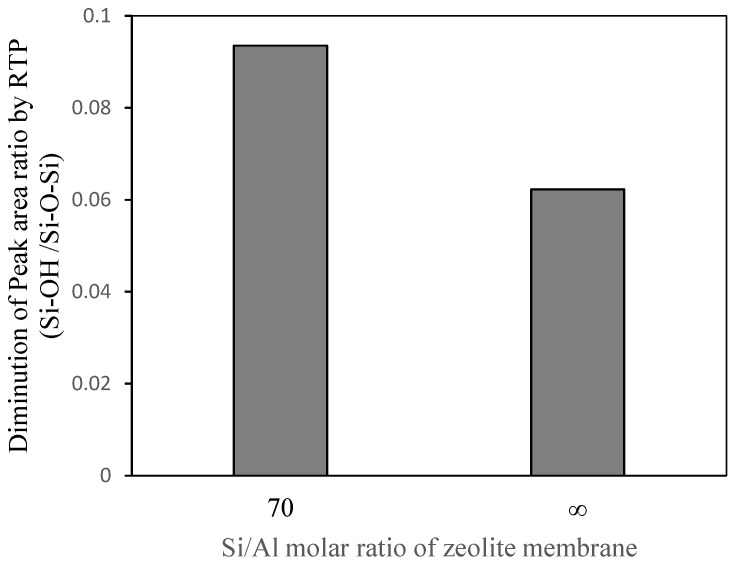
Diminution of the peak area ratio (Si–O–H/Si–O–Si) of ZSM-58 membranes with different Si/Al molar ratios after RTP treatment.

**Table 1 membranes-11-00623-t001:** Si/Al molar ratios of ZSM-58 crystals at various gel compositions at 423 K for 24 h.

Crystal No.	Gel Molar Ratio				Me Source *	Measurement of Si/Al with XRF
	MTI/SiO_2_	MeOH/SiO_2_	Si/Al	H_2_O/SiO_2_		
C1	0.05	0.05	70	52	NaOH	116
C2	0.1	0.1				112
C3	0.1	0.3				82
C4	0.3	0.1				91
C5	0.05	0.05	70	52	KOH	79
C6	0.1	0.1				72
C7	0.1	0.3				52
C8	0.3	0.1				73

* Me: alkali metal cation source.

**Table 2 membranes-11-00623-t002:** Equimolar CO_2_/CH_4_ mixture separation performances of Al-containing ZSM-58 zeolite membranes with different synthesis times at a total feed pressure of 1.1 MPa and at room temperature.

No.	Synthesis Time (h)	Thickness(μm)	CH_4_ Permeance of the Noncalcined Membrane * (×10^−10^ mol/m^2^/s/Pa)	Permeance (mol/m^2^/s/Pa)		CO_2_/CH_4_ Selectivity
				×10^−8^ PCO_2_	×10^−10^ PCH_4_	
M1	72	4.1	3.1	11	87	13
M2	84	8.5	1.8	8.5	29	29
M3	96	12	0.8	6.1	13	45
M4	120	8.0	<0.1	7.8	5.3	150

* CH_4_ leak permeance was measured before calcination.

**Table 3 membranes-11-00623-t003:** Equimolar CO_2_/CH_4_ mixture separation performances of all-silica ZSM-58 zeolite membranes under a total feed pressure of 1.1 MPa at room temperature.

No.	Calcination Method	Si/Al	Thickness(μm)	Weight Loss (%)	CH_4_ Permeance of Noncalcined Membrane * (×10^−10^ mol/m^2^/s/Pa)	Permeance (mol/s/m^2^/Pa)		CO_2_/CH_4_ Selectivity
						×10^−8^ PCO_2_	×10^−10^ PCH_4_	
M5	RTP + CTC	∞	3.3	11	0.3	13	520	2.5
M6	Ozone	∞	3.4	6.6	6.3	8.0	5.5	150

* CH_4_ permeance was measured before calcination.

**Table 4 membranes-11-00623-t004:** Synthesis conditions and peak area ratios of silanol concentrations of ZSM-58 crystals with different Si/Al molar ratios with the measured FTIR spectra.

Si/Al	MTI/SiO_2_	KOH/SiO_2_	H_2_O/SiO_2_	Time (h)	Temperature (K)	Peak Area Ratio (Si–OH/Si–O–Si)	
						Before RTP	After RTP
40	0.25	0.33	40	96	433	0.082	0.010
70	0.05	0.05	52	120	413	0.083	0.022
300	0.1	0.1	52	72	413	0.071	0.040
∞	0.1	0.1	52	48	413	0.065	0.034

**Table 5 membranes-11-00623-t005:** Peak area ratios of the silanol concentration of the ZSM-58 membrane with different Si/Al molar ratios measured by FTIR.

Si/Al	Peak Area Ratio (Si–OH/Si–O–Si)	
	Before RTP	After RTP
∞	0.149	0.086
70	0.162	0.069

## Data Availability

Data available upon request.

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
