# Peer review of "Preparation of Al-Containing ZSM-58 Zeolite Membranes Using Rapid Thermal Processing for CO2/CH4 Mixture Separation"

_membranes, 2021, doi:10.3390/membranes11080623_

Round 1

Reviewer 1 Report

The manuscript showed an approach to achieving efficient template removal and crack-free Al-containing ZSM-58 membranes, and the effect of RTP treatment by confirming reaction of silanol gropes in ZSM-58 zeolite by FT-IR were demonstrated systematically. The very good effect of the new membrane one CO2 selectivity has been shown by the figures and tables. It is very interesting. It is suggested authors should compare the new membrane with previous membranes in detail so that readers could know how effective the new approach is. In addition, the conclusion is too long.

Reviewer 2 Report

Reviewer comments

This work describes the synthesis and testing of Al-containing ZSM-58 molecular sieve membranes grown on alumina tubular supports. The authors studied the effect of MTI/SiO2 and MeOH/SiO2 ratios and alkali source on the final Si/Al ratio composition in the membranes and found that KOH was a better alkali source than NaOH. The authors also determined the best crystal growing conditions (reactants concentrations, temperature, time) for membrane fabrication. Post synthesis treatment of the membranes included a calcination step with a combination of the RTP and CTC protocols that removed impurities from the membranes.  Gas permeation data was used to determine the best membrane growth period from the gel in the hydrothermal treatment.

General comments:

  • The article needs a thorough English grammar revision. In its present form, the reading is not fluid and makes it difficult to follow the line of though and to understand the conclusions.
  • It is difficult to follow the sequence and purpose of the experiments the authors performed. It would be helpful to the reader if the authors include a brief summary of the sequence of experiments and their purpose or goal at the end of the introduction section to guide the reader.

Sections comments:

  1. Introduction:

Include the thermal expansion coefficients for both materials in “…but this process generates cracks because of the differences in the thermal expansion coefficient between the zeolite and porous substrate [8,9].” to illustrate the difference.

“The ozone de-template method can remove SDAs at lower temperatures than the conventional de-template method by ozone with high oxidizability”  This sentence is not clear, may need to be rephrased and explained. What is the difference between the “ozone de-template method” and the “conventional de-template method by ozone”?

  1. Experimental procedure

Methylpropiniumiodide (MTI) is not listed in the materials section.

“Table 1 Figure 1 shows the properties and scanning electron microscope (SEM) image of the alumina support, respectively.”  Table 1 and Figure 1 maybe? The sentence is incomplete.

Figure 1: Include copyright information and reference for this figure.  Figure 1 was published in reference 5: E. Hayakawa, S. Himeno, Synthesis of all-silica ZSM-58 zeolite membranes for separation of CO2/CH4 and CO2/N2 gas mixtures, Microporous Mesoporous Mater. 291 (2020) 109695.

In the paragraph “… porous α-alumina tubes (pore size, 200 nm; outer diameter, 10 mm; inner diameter, 7 mm; length, 3 cm; NGK Insulators, Ltd.).” and in Table 1, the support’s thickness should be the difference between the outer and inner radius, that is 1500 micrometers or 1.5 mm but in the table is 1050 micrometers, which one is correct?

The wording and phrases in section 2.2 "Preparation of ZSM-58 seed crystals” are incredibly similar to the wording in section 2.2 of ref. 5.  Rephrase this section.

Figure 2a): SEM image is a cropped copy of Figure 2 from ref. 5, again get copyright permission from Elsevier and ref. 5 in the figure description, or use an unpublished SEM image.

Figure 2b) The experimental XRD for ZSM-58 seed crystals is exactly the same as in Figure 3 of ref. 5.  Replace it with one that was not published before.

Section 2.3

“…Teflon-lined autoclave, and hydrothermal treatment was conducted at constant temperature and time.” Constant time? Revise this sentence.  What temperature? For how long?

“…Finally, the ZSM-58 zeolite crystals were calcined at 823 K for 4 h at a ramp rate of 0.5 K/min.” Calcined in vacuum or inert atmosphere?

Reference Table 2 in section 2.3 to relate the synthesis conditions for C1-C8.

Section 2.6: In the materials section the authors listed CO2 (99%) and CH4 (99%) gases for permeability studies but they do not mention the CO2/CH4 blend.  Given that a CO2/CH4 blend was used for gas permeation, at the experimental pressure of 10 atm a stage cut maybe mandatory to minimize concentration polarization. Concentration polarization in gas mixture separation affects adversely the selectivity of membranes since the surface of the membrane is not exposed to a constant feed composition during the separation process, the retentate must be removed at rate fast enough to ensure the composition at the surface of the membrane is constant (the equimolar blend of CO2 and CH4).  Was a stage cut used in the permeability experiments?

  1. Results and discussion

C8 is not mentioned in Figure 6’s caption.

What was the calcination atmosphere used for M1-M4?

Table 3:  The authors measured the thickness of the membranes deposited on the tubular supports, what is the standard deviation of these measurements?  If thickness was measured, the authors should be able to estimate the permeability of the membranes and include them in this table.  Is the data for just one membrane of each type of membrane? If it is just one membrane then the data may not be reproducible.  The authors should make the effort to duplicate their permeability results and provide and average of permeance and selectivity data of at least 2 membranes.

How do the authors explain that M4 has smaller thickness than the M1-M3 membranes if it was grown for a longer time?

What is the difference between the CH4 leak permeance and the CH4 permeance measurements? This may need to be clarified in the text.  If the authors are trying to determine the presence of cracks in the membranes, then propane (0.43 nm) or propylene (0.45 nm) would be better probes than methane (0.38 nm).

“The CH4 permeance of the membrane before calcination was very low, and it suggested that the dense zeolite membrane was formed by C5 composition. In addition, the density of the zeolite membrane was higher as the synthesis time was longer, observed from the CH4 leak test and CH4 permeance.”    It is obvious that the permeance of the membrane before calcination is going to be low due to impurities and adsorbed water in the pores of the zeolite, these impurities are removed at high temperatures, e.g., zeolite A desorbs water at temperatures above 300 °C at atmospheric pressure.  How is the density of the membrane related to the CH4 leak test and the CH4 permeance? This claim needs to be clarified and supported with references in the text.  A more plausible explanation would be that there are less defects on the membrane at longer growing times than at shorter growing periods.

Section 3.3

This entire section needs a major rephrasing and English grammar checking to understand the analysis the authors made. As it is written, it is difficult to understand and to follow what the authors conclude.  The authors should also include the thicknesses in numbers of each layer they show in Figure 9a,b,c.

Section 3.4

Comparing the separation performances of M4 and M5 is not correct. M4 was grown in a period of 120 hours and M5 in a period of 48 hours.  From Table 3 it is clear that 72 hours is not enough to provide a non-defective layer, therefore a more scientifically objective comparison would require an M5 membrane grown for 120 hours with a similar thickness to that of M4 to ensure the RTP+CTC protocol applied to Al-containing ZSM-58 produces more robust membranes than those from ZSM-58 membranes.  Actually, a fairer comparison would be that of M1 with M5 because they have similar thicknesses and growing times, not to mention similar selectivity to M5’s.

This section also needs of major rephrasing and English grammar corrections.

Section 3.5

The reading of this section is not fluid and makes it difficult to follow.  This section needs a rephrasing and English grammar checking.

  1. Conclusions

Cannot comment on the authors’ conclusions because sections 3.3 to 3.5 are difficult to read and follow.

Round 2

Reviewer 2 Report

1) Tables 3 and 4:  Check permeance exponents for gases, they should be 10-8 or 10-10 for CO2 and CH4, otherwise CH4 permeance looks larger than CO2's.

2) English still needs a thorough grammar and spelling check.

3) The authors should perform a careful reading of the text to correct for typos and missing characters.

Author Response

Dear reviewer

We thank referees for careful reading our manuscript and for giving useful comments. In response to the reviewer comments, we have revised the membranes-1304353.

We have corrected unit notation in Table 3 and Table 4.

Moreover, we had it edited by a native English speaker.

We look forward to a publication of our manuscript in special issue “Membranes for Gas Separation and Purification Processes” of Membrane.